# Multivariate characterisation of a blackberry-alder agroforestry system in South Africa: Hydrological, pedological, dendrological and meteorological measurements

Sibylle Kathrin Hassler[1,2], Rafael Bohn Reckziegel[3], Ben du Toit[4], Svenja Hoffmeister[1], Florian Kestel[5], Anton Kunneke[4], Rebekka Maier[6], and Jonathan Paul Sheppard[3]

[1]Karlsruhe Institute of Technology (KIT), Institute of Water and Environment - Hydrology, Karlsruhe, Germany
[2]Karlsruhe Institute of Technology (KIT), Institute of Meteorology and Climate Research, Atmospheric Trace Gases and Remote Sensing, Karlsruhe, Germany
[3]University of Freiburg, Faculty of Environment and Natural Resources, Chair of Forest Growth and Dendroecology, Freiburg im Breisgau, Germany
[4]Stellenbosch University, Department of Forest and Wood Science, Stellenbosch, South Africa
[5]Leibniz Centre for Agricultural Landscape Research (ZALF) e.V., Research Area 1 "Landscape Functioning", Müncheberg, Germany
[6]University of Freiburg, Faculty of Environment and Natural Resources, Chair of Soil Ecology, Freiburg im Breisgau, Germany

**Correspondence:** Sibylle K. Hassler (sibylle.hassler@kit.edu)

**Abstract.** Trees established in linear formations can be utilised as windbreak structures on farms as a form of agroforestry system. We present an extensive data package collected from an active berry farm located near Stellenbosch, South Africa considering hydrological, pedological, dendrological and meteorological measurements centred around an Italian alder (*Alnus cordata* (Loisel.) Duby) windbreak and a blackberry (*Rubus fruticosus* L. Var. 'Waldo') crop. Data was collected between September 2019 and June 2021. The data is available from Hassler et al. (2024) and includes: solar radiation, precipitation characteristics, vapour pressure deficit, air temperature, humidity, atmospheric pressure, wind speed and direction, gust speed, lightning strikes and distance recorded at 10 minute intervals; soil moisture and matric potential in two profiles at 15 minute intervals alongside soil samples at various depths describing soil texture, hydraulic conductivity and water retention parameters; a soil profile description accompanied by 60 topsoil samples describing carbon and nitrogen and exchangeable base cation concentrations as well as potential cation exchange capacity and descriptions of soil texture; dendrological measurements present point cloud data for the study windbreak trees and surrounding features alongside cylinder models of the windbreak trees with volume and biomass data, foliage data as a product of an existing leaf creation algorithm is also given. The described dataset provides a multidisciplinary approach to assess the impact and interaction of windbreaks and tree structures in agroforestry landscapes, aiding future work concerning water fluxes, nutrient distribution, microclimate, and carbon sequestration. The dataset, including high-resolution time series and point cloud data, offers valuable insights for managing the windbreak's influence and serves as a unique training dataset for spatial analysis.

# 1 Introduction

Agroforestry systems (AFS) are deliberate and targeted combinations of agriculture with forestry established within the same land management unit. AFS present a broad range of systems for the co-production of timber and fruits with crops and/or livestock alongside multiple other benefits and services. AFS are flexible systems which can be adjusted to suit the land manager's needs and can be considered an optimisation of space where a multitude of products can be cultivated on different spatial and temporal levels.

AFS have the potential to provide a suite of social, economic and environmental benefits, presenting a far more stable and long-term solution for provisioning, protective and buffering needs when compared, for example, with arable monoculture or livestock rearing (Sheppard et al., 2020). Of increasing importance is the acceptance that employing AFS offers the potential to preserve and protect natural resources and biodiversity against the effects of climate change influences, as in many parts of the world drought and extreme climatic events are expected to become more frequent (New et al., 2006; Meadows, 2006; IPCC, 2014). South Africa, particularly, is expected to experience shifting rainfall patterns and increased temperatures, therefore, climate change adaptation strategies, the implementation of AFS being one particular tool in the toolbox, are of the highest importance to mitigate future pressures (Veste et al., 2024; Ziervogel et al., 2014).

By applying AFS, we have a possibility of enhancing food security, improving the rural bio-economy and providing an adaptation strategy for human needs with a more natural resource management approach which is considered important, particularly in South Africa (Sheppard et al., 2020; Zerihun, 2021). The concept of land equivalent ratio (LER) (for example as described by Ong and Kho, 2015) is a metric that evaluates the yield of agroforestry crops in direct comparison with monoculture cropping systems. By applying the LER calculation it is demonstrated that AFS have the potential to provide greater yields on an equivalent land area. Likewise, the inclusion of woody biomass also offers the possibility of either long or short-term carbon sequestration or in the form of carbon substitution by providing an alternative fuel supply. These are all factors that have profound social and economic implications, in the present and near future, especially under a changing climate. Additionally, the environmental benefits brought about by the establishment of AFS in farmed and formally tree-less environments are multiple (Sheppard et al., 2020), especially at a local scale. For example, soil nutrient removal by soil erosion processes is known as a major cause of land degradation (Montanarella et al., 2016) and affects agricultural productivity, and thus, livelihood and food security, especially in areas with a high dependency on subsistence farming. The implementation of AFS practices provides a protective function and has been shown to decrease soil erosion by stabilising the soil and disrupting erosive wind and run-off events (Mbow et al., 2014). The inclusion of trees in previously tree-less environments can positively increase the rate of nutrient cycling or boost storage processes (Hombegowda et al., 2016). The presence of trees can also influence water infiltration and redistribution between soil layers (Burgess et al., 1998; Anderson et al., 2009). Moreover, trees and woody perennials can be used as a source of nutrient input either through nitrogen fixation or by the manual incorporation of biomass in productive fields. Trees in AFS may also affect the microclimate of the site, and also, of the crops. For example, the use of trees as a linear windbreak provides a reduction in near ground wind speed modifying both air temperature and air humidity, thus, positively influencing soil properties, crop growth and yield (Brenner et al., 1995). Moreover, an increase in the structural diversity of

farmed land will increase the overall biodiversity and biodiversity services (such as integrated pest management). Nevertheless, both actual and perceived negative effects of the introduction of trees in agricultural fields are also perceptible. These may include issues such as shading, a loss of productive area or an increase of obstacles in the field that hinder management and are dependent on individual production goals, location and site-specific characteristics. Research and modelling efforts are, for example, making inroads into this field of study e.g. the management of shading effects of agroforestry trees by Bohn Reckziegel et al. (2021; 2022).

The positive and negative effects of the utilisation of trees within a farmed landscape are multi-faceted and dependent on the combination between tree and crop (or tree, crop and/or livestock). They can be examined with a multi-dimensional and trans-disciplinary approach, studying soil mechanical processes, nutrient cycling, hydrological fluxes, shifts in micrometeorologic regime, as well as assessing the carbon sequestration potential. Thus, by studying functional combinations of AFS and by applying modifications and trade-offs to to traditional agricultural methods, best management practices and decision support systems for the successful inclusion of trees in previously tree-less environments can be developed.

We conducted an inter-disciplinary field campaign in an AFS on an active berry farm in South Africa, to study the effects of trees in the site conditions, focusing on the water fluxes, the soil nutrients and the microclimate. The campaign concentrated on a system combining blackberry production and an established windbreak consisting of alder trees. The scrutiny of hydrological, pedological, dendrological and meteorological measurements on one study site as presented here, allows for a more objective viewpoint on the processes and influences that are occurring.

## 2    Site description

The field site is situated on the southeast facing flank of the Simonsberg near Stellenbosch, in the Western Cape province in South Africa. The region's climate is classified as Mediterranean with hot, dry summers (Dec-Mar) and cool, wet winters (May-Sep) (Meadows, 2015). The wind direction in the area switches from predominantly northwest in the winter months to south easterly winds in the summer months, but experiences annual variations in this trend (Veste et al., 2020).

The geological setting is formed by granites, quartzporphyry and syenite (Schifano et al., 1970) from which a Dystric Cambisol developed (IUSS Working Group WRB, 2015). The field site is situated at roughly 400 m above mean sea level and the slope is inclined by approximatly 15 %.

The site is part of a working fruit farm and the studied field is under current cultivation with blackberry (*Rubus fruticosus* L. Var. 'Waldo') divided by linear windbreaks consisting of pure Italian alder (*Alnus cordata* (Loisel.) Duby). The blackberry canes are aligned in parallel rows with a between-row distance of 2.5 m. At the time of study the blackberry canes at the site were 5-6 years old, these usually start shooting in late spring (October) and the berries are harvested from mid-January to mid-March. The plants are irrigated with a dripper system during summer (Nov-Jan), roughly three times per week for 1.5 hours at a rate of 2.3 L hr$^{-1}$. Slow release fertiliser is applied once a year just before spring.

Our measurement campaign focused on two berry plots and the adjacent alder windbreaks. The windbreak in between the berry plots was studied in more detail. It consists of one single row of alder trees 45 m long, approximately 15 to 20 years old,

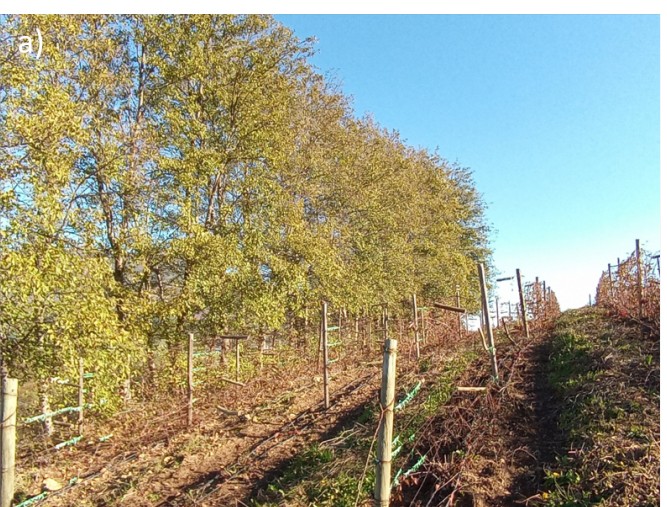 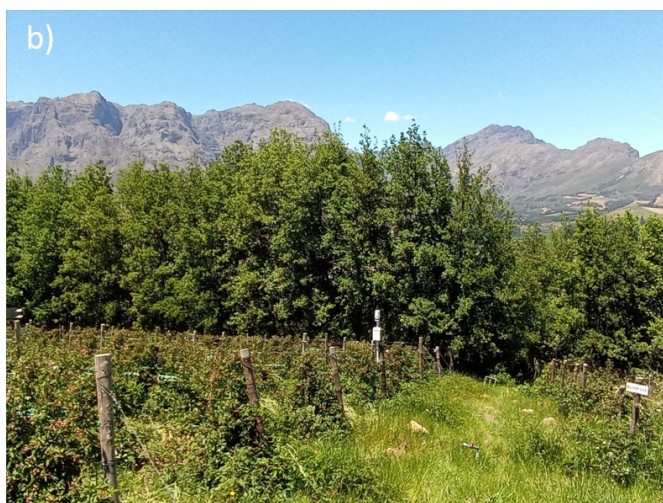

**Figure 1.** Images from the field site taken in a) winter and b) late spring. ©Anton Kunneke

and is linearly oriented from east-northeast to west-northwest. Individual trees are planted at a regular spacing of approximately
70 cm. The trees developed very particular crown shapes, perpendicular to the windbreak axis, with exceptions of edge trees
which have a rounded crown. Impressions of the site and the differences in the seasonal appearance are shown in Figure 1.

## 3 Measurement setup

A joint interdisciplinary field campaign took place in September 2019. This activity included the extraction of soil samples
for texture determination and nutrient and soil organic carbon analysis, along transects across the two berry plots and adjacent
windbreak. Additional sampling with larger cylinders was done to determine soil hydraulic characteristics. To characterise
and classify the soil, a profile was dug, described and sampled. Scanning the windbreak between the two berry plots with
terrestrial LiDAR was coupled with manual measurement of the windbreak trees. Additionally, various sensors were installed
to continuously monitor meteorological and soil hydrological variables in the subsequent months. A map of the study site is
shown in Figure 2. All measurements are described in more detail in the following sections.

### 3.1 Meteorological variables

#### 3.1.1 Measurements

Meteorological data was recorded with an ATMOS41 weather station in combination with a ZL6 Cloud Data Logger (both
METER Group Inc., Pullman, USA). Figure 2 shows the position of the weather station on the study site. A list of the measured
variables and their sensors along with range, resolution and accuracy is compiled in Table 1. A rough overview of the resulting
time series is attached in the Appendix (A1).

**Table 1.** Sensor characteristics for the meteorological measurements. All sensors are part of the setup of the ATMOS41 weather station (METER Group Inc., Pullmann, USA).

| Measurement | Instrument | Company | Range | Resolution | Accuracy |
|---|---|---|---|---|---|
| Solar radiation | pyranometer | Apogee Instruments | 0–1750 W m$^{-2}$ | 1 W m$^{-2}$ | ±5 % |
| Precipitation | optical sensor rain gauge | METER Group Inc. | 68 cm$^{-2}$ catch area | 0.017 mm | ±5 % |
| Maximum precip rate | optical sensor rain gauge | METER Group Inc. | 0–400 mm h$^{-1}$ | 1 W m$^{-2}$ | ±5 % |
| Air temperature | thermistor | METER Group Inc. | −50 to 60 °C | 0.1 °C | ±0.6 °C |
| Relative humidity | relative humidity sensor | METER Group Inc. | 0–100 % (0.00–1.00) | 0.1 % | ±3 % |
| Temp. of humidity sensor | thermistor | METER Group Inc. | −40 to 50 °C | 0.1 °C | ±1.0 °C % |
| Atmospheric pressure | barometric pressure sensor | METER Group Inc. | 50–110 kPa | 0.01 kPa | ±0.1 kPa from −10 to 50 °C and ±10.5 kPa from −40 to 60 °C |
| Vapour pressure deficit | barometric pressure sensor | METER Group Inc. | 0–47 kPa | 0.01 kPa | ±0.2 kPa below 40 °C |
| Wind direction | ultrasonic anemometer | METER Group Inc. | 0°–359° | 1° | ±1° |
| Horizontal wind speed | ultrasonic anemometer | METER Group Inc. | 0–30 m s$^{-1}$ | 0.01 m s$^{-1}$ | 0.3 m s$^{-1}$ |
| Wind gust speed | ultrasonic anemometer | METER Group Inc. | 0–30 m s$^{-1}$ | 0.01 m s$^{-1}$ | 0.3 m s$^{-1}$ |
| Lightning strike count | radio wave sensor | METER Group Inc. | 0–65535 strikes | 1 strike | >25 % detection at <10 km |
| Lightning average distance | radio wave sensor | METER Group Inc. | 0–40 km | 3 km | - |

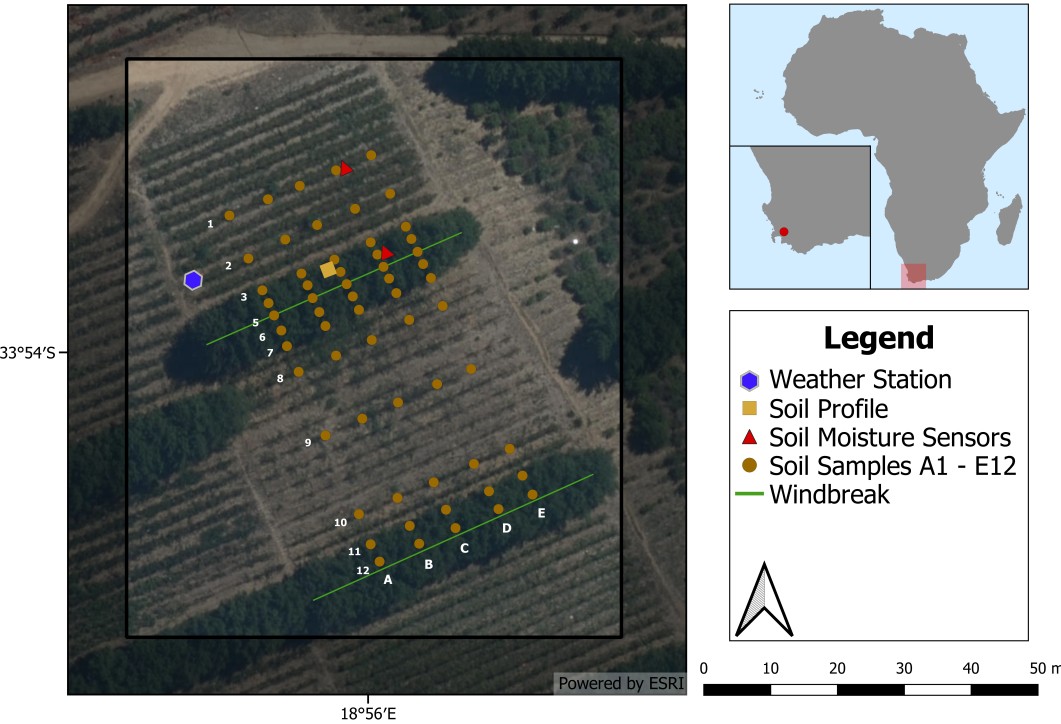

**Figure 2.** Map of the study site showing the location of the measurements of the field campaign (from the data description to Hassler et al., 2024).

### 3.1.2 Data processing and quality control

Data was recorded in 10-minute intervals from 19 September 2019 until 18 June 2021. The time zone utilised is Coordinated Universal Time (UTC). Sensors were installed two metres above ground level. During high precipitation events the ultrasonic anemometer recorded unusually high values. This error was on occasion also experienced in the mornings, probably due to dew accumulation. All events in question were referenced with the climate station at Stellenbosch airport, and wind speed as well as gust speed were manually filtered as NA, whenever values seemed unreasonable. The procedure was straightforward, as most of the unusual values exceeded the maximum possible $30$ m s$^{-1}$ of the sensor on low wind days. It should be taken into account that wind speeds are significantly reduced by the windbreak plantings when the wind is blowing in a southerly direction, which leads to considerable differences in wind speeds when wind direction changes frequently in short periods of time. The accuracy of the humidity sensor is generally slightly lower in a high humidity environment as it is a temperature-dependent variable that shows large fluctuations with small changes in temperature, the higher the number of vapour particles detected. The ATMOS41 weather station measures precipitation in individual droplets to reflect also dew formation with high

accuracy and queries appropriate accumulations once every minute in memory. Precipitation is then given as cumulative amount according to a chosen interval and as a maximum rate for the respective interval multiplied by interval/60 min h$^{-1}$.

## 3.2 Hydrological measurements

We studied the influence of the windbreak on water fluxes in the berry plot by monitoring soil moisture and matric potential. Additionally, we took soil samples to characterise soil hydraulic characteristics under laboratory conditions.

### 3.2.1 Soil moisture and matric potential

Soil moisture was monitored continuously with time domain reflectometry (TDR) probes (Pico-profile T3PN, IMKO GmbH, Ettlingen, Germany) at two locations in four different depths (Figure 3), from 21 September 2019 to 14 March 2020. The probes were installed in acrylic glass access tubes (diameter of 4.4 cm). One tube was installed within the rhizosphere of the windbreak, in the first berry row (referred to as "windbreak + berries" in the following). The other tube was installed close to the 8th berry row from the windbreak ("berries"; reference location), to minimise its influence. In each tube, we installed four TDR probe segments, stacked directly on top of each other to allow for a continuous soil moisture signal along the profile. Each probe segment had a length of 18 cm and a penetration depth of the microwave impulse of 5.5 cm, so the soil moisture signal integrated over a volume of approximately 1 dm$^3$. The total profile depth we could measure was subsequently about 0.8 m, which should cover a prominent influence zone of the tree roots in the "windbreak + berries" tube (Kutschera and Lichtenegger, 2002), allowing comparisons with the reference location.

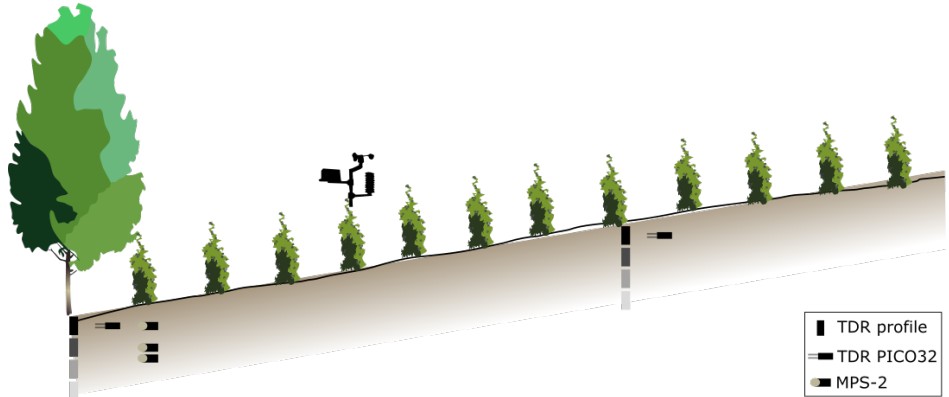

**Figure 3.** Slope cross-section sketch with the monitoring setup to study the water fluxes. The soil samples for measuring soil hydraulic characteristics in the laboratory are not included in the sketch; the profile samples were taken adjacent to the monitoring setup close to the windbreak, and the 12 extra samples directly next to the windbreak and in the reference berry row.

Additionally, we installed two TDR probes (Trime PICO32, IMKO GmbH, Ettlingen, Germany) with a measurement volume of approximately 0.25 dm$^3$, one next to each tube, at 10 cm depth to capture the likely more dynamic soil moisture changes close to the surface (Figure 3). Adjacent to the "windbreak + berries" tube location, we also installed three dielectric water

**Table 2.** Sensor characteristics for the hydrological measurements as given by the manufacturer.

| Measurement | Instrument | Company | Range | Resolution | Accuracy |
|---|---|---|---|---|---|
| Soil moisture | PICO-T3PN | IMKO GmbH | 0–100 % | - | $\pm 2$ % |
| Soil moisture | TRIME PICO32 | IMKO GmbH | 0–100 % - | - | $\pm 2$ % (0–40 % VWC), $\pm 3$ % (40–70 % VWC) |
| Matric potential | MPS-2 | Decagon Devices Inc. | $-5$ to $-500$ kPa | 0.1 kPa | $\pm 2$ % of reading from $-5$ kPa to $-100$ kPa |
| Soil temperature | MPS-2 | Decagon Devices Inc. | -40 °C to +50 °C | 0.1 °C | $\pm 1$ °C |

potential sensors (MPS-2, Decagon Devices Inc., Pullman, WA, USA) in a profile at depths of 0.1, 0.3 and 0.4 m (Figure 3) to gain more insight into the water tension in the soil in different depths under the influence of the tree roots. The monitoring measurements were recorded with a Truebner TrueLog 100 (Truebner GmbH, Neustadt, Germany) data logger in 15-min intervals. An overview of the sensors and their range, resolution and accuracies can be found in Table 2, a figure of the resulting measurement time series is included as Figure 4.

### 3.2.2 Soil hydraulic characteristics

Undisturbed 250 ml cylindrical soil samples were taken to measure soil hydraulic characteristics in the laboratory. Three samples were extracted from a soil profile adjacent to the soil hydrological measurements at the windbreak (Figure 3). The samples were taken at the surface and at depths of 0.3 and 0.5 m right after installation of the sensors on 16 September 2019. In a sampling campaign in March 2022, twelve additional samples were taken to enable a more comprehensive assessment. They were located at three points (east, middle and west) directly adjacent to the windbreak, and at three points within the reference berry row (eighth row from the windbreak), always at 5 and 25 cm depth.

Soil texture analysis was conducted through wet sieving (without soil skeleton of particle diameter $d_p > 2$ mm) into the classes of coarse sand (2 mm $> d_p > 630$ $\mu$m), medium sand (630 $\mu$m $> d_p > 200$ $\mu$m) and fine sand (200 $\mu$m $> d_p > 63$ $\mu$m). Before, organic compounds were destroyed through the application of hydrogen peroxide. Smaller fractions were separated with the sedimentation method according to DIN ISO 11277 (ISO, 2002) into coarse silt (63 $\mu$m $> d_p > 20$ $\mu$m), medium silt (20$\mu$m $> d_p > 6.3$ $\mu$m), fine silt (6.3 $\mu$m $> d_p > 2$ $\mu$m) and clay ($d_p < 2$ $\mu$m).

Soil hydraulic conductivity of the undisturbed samples was measured with the Ksat apparatus (UMS GmbH, Munich). The measurement is following the Darcy approach applying water flux through a saturated porous medium. The apparatus records the falling head of the water supply though a highly sensitive pressure transducer which is used to calculate the flux.

Soil water retention characteristics were measured on the same samples in the HYPROP apparatus (UMS GmbH, München, Germany) and afterwards in the WP4C PotentiaMeter (Decagon Devices Inc., Pullman, WA, USA). The HYPROP registers matric head in two depths and the total weight of the sample while it is drying out by free evaporation. Maximum water tensions that can be measured with this method are about $-800$ hPa and are limited by the air entry point of the tensiometer.

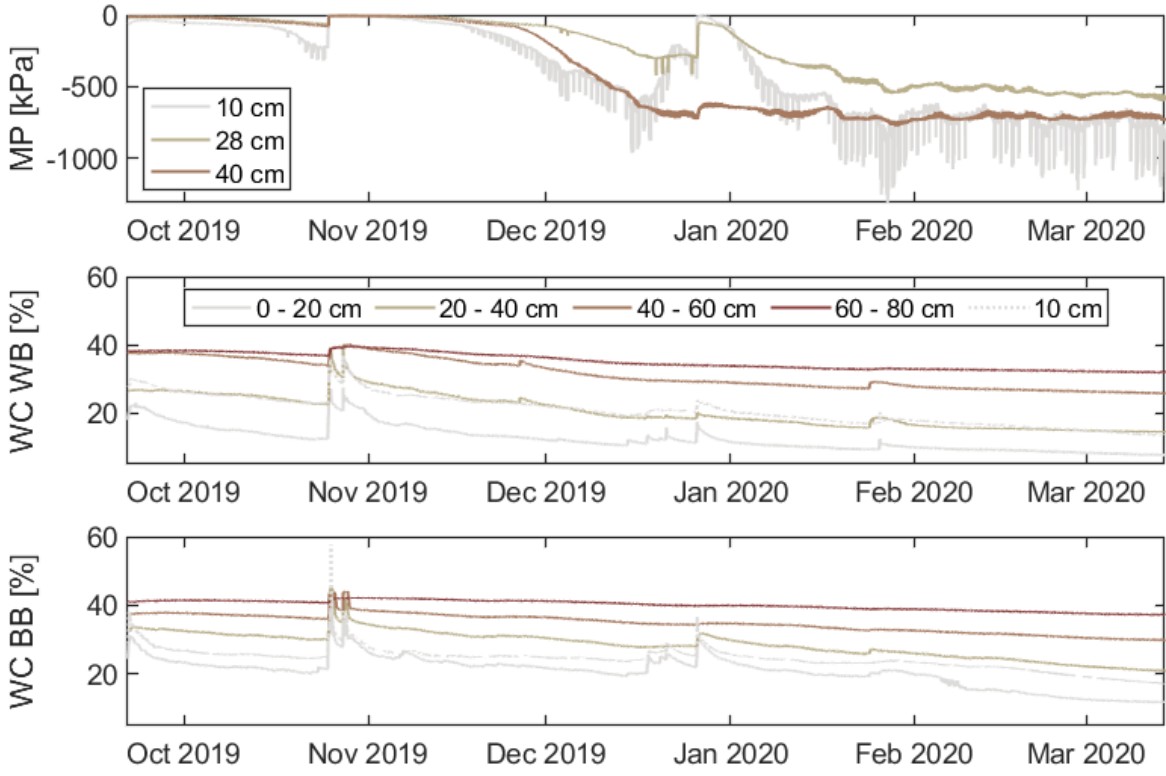

**Figure 4.** Time series of soil moisture and matric potential over the course of the monitoring period. WC: water content, MP: matric potential, WB: "windbreak + berries" measuring location, BB: reference location "berries".

After measurement in the HYPROP, a small subsample of about 10 g was transferred to the WP4C where soil water potential is measured using a chilled mirror approach.

The measurements of water tension and water content from the HYPROP and WP4C were used to derive water retention curves (Figure 5), parameterised with the van Genuchten equation (van Genuchten, 1980):

$$\Theta(\psi) = \Theta_r + \frac{\Theta_s - \Theta_r}{[1 + (\alpha|\psi|)^n]^{1-1/n}} \tag{1}$$

with the water retention curve $\Theta(\psi)$ in m$^3$ m$^{-3}$, the residual water content $\Theta_r$ in m$^3$ m$^{-3}$, the saturated water content $\Theta_s$ in m$^3$ m$^{-3}$, the scaling parameter inverse to the air entry suction $\alpha$, the matric potential $\psi$ and the dimensionless shape parameter $n$.

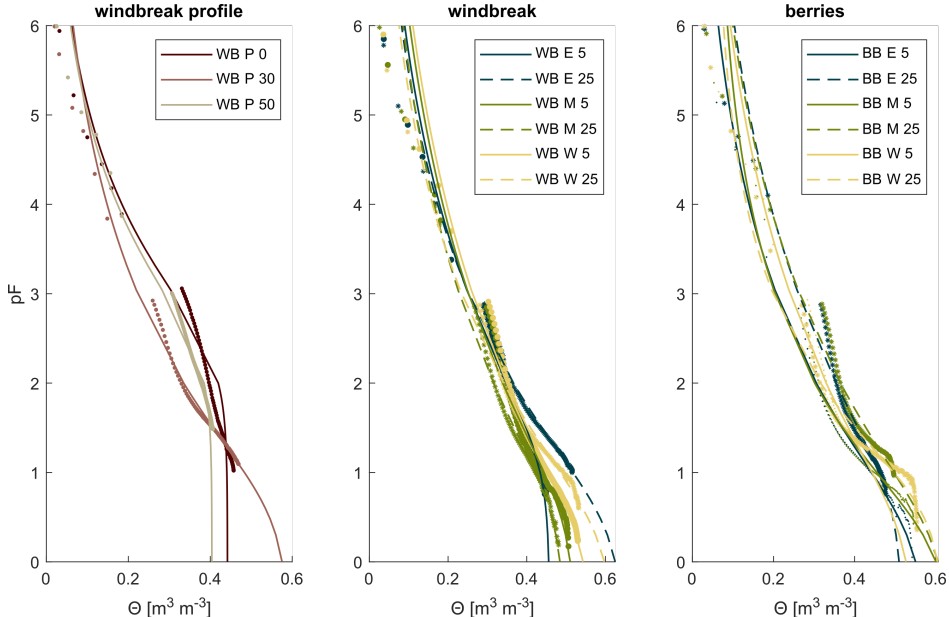

**Figure 5.** Soil water retention curves of the three profile samples at the windbreak (left panel, at 0, 0.3 and 0.5 m depth), six samples at three locations at the windbreak (middle panel, at 0.05 and 0.25 m depth) and six samples at three locations in the berries (right panel, at 0.05 and 0.25 m depth). The dots/stars represent the pF and water content measurements, the lines show the fitted water retention curves after van Genuchten (1980). In the middle and right panel, dots belong to the 0.05 m depths, stars to the 0.25 m depth.

### 3.2.3 Data processing and quality control

The in-situ monitoring sensors recorded observations in 15-minute intervals. The data were checked for inconsistencies, obvious erroneous values and spikes or missing values. These could occur due to power outages or maintenance interruptions. The respective values were removed from the time series. The time stamp which was originally in local time was converted to the UTC time format for all time series.

An accuracy of 2 % can be reached with the IMKO sensors (T3PN and PICO32) if the tubes/sensors are installed correctly
with close contact to the soil. No site-specific calibration was done prior to or post installation, which can lead to errors in the absolute readings. Further, we did not correct for soil density deviations but used the standard soil calibration (1.40675 g cm$^{-3}$).

The MPS-2 matric potential sensors frequently reached values of less than $-500$ kPa, which should not be used for quantitative analyses because of potentially large errors in the magnitude of the water potential.

It is plausible that some degree of error could have been introduced during installation of the sensors. However, drilling the vertical holes for the acrylic tubes was done with great care so that the tubes had good contact with the soil. Additionally, any space between the soil and the tube could be assumed to settle with time and with rain so that the measurement signal should be robust. Due to heterogeneities in the soil, the additional Trime PICO TDR sensors and the matric potential sensors may not

have been installed within completely uniform soil conditions as a result of air pockets or rocks. This problem is difficult to
avoid completely, however, during installation we did not observe any gross heterogeneities that would indicate concern.

When examining water fluxes and calculating water balances, the factor of irrigation should be kept in mind, as the additional
water input cannot be exactly quantified.

## 3.3    Soil characteristics

Soil sampling took place during the joint field campaign in September 2019. We collected samples from a soil profile for soil
classification and along transects to study the spatial differences in topsoil nutrient contents and texture (see Figure 2).

### 3.3.1    Soil profile

In a freshly dug soil profile to a depth of 0.75 m we defined soil horizons after visual inspection of colour changes and manual
texture assessment (Figure 6). The horizons were: Ap: 0-20 cm; AhB: 20-40 cm; Bw1: 40-55 cm; Bw2: 55-75 cm; C: >75
cm. From each horizon, we took one composite sample mixed from multiple individual ones, spread across the width of the
soil pit and the depth of the respective horizon, using a hand shovel. We used these samples for texture analysis and determi-
nation of total carbon and nitrogen as well as the potential cation exchange capacity (CECpot) and contents of exchangeable
cations. These were carried out under laboratory conditions in Germany. A soil profile description and soil classification was
subsequently carried out according to the World Reference Base for Soil Resources (IUSS Working Group WRB, 2015).

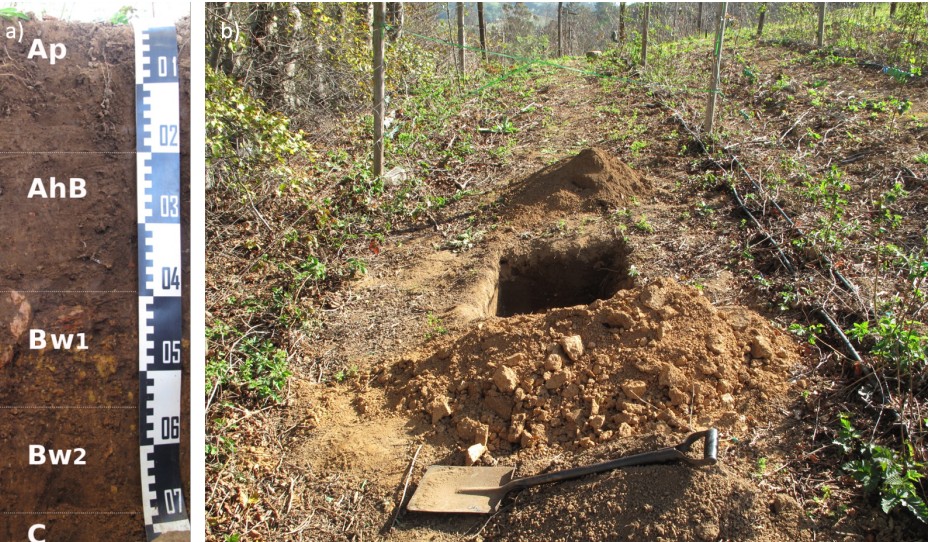

**Figure 6.** Soil profile: Dystric Cambisol (loamic, colluvic, humic) according to the IUSS Working Group WRB (2015). a) Profile with the
delineated horizons, b) profile pit in between the berry rows.

### 3.3.2   Transects

We further collected 60 soil samples along five parallel transects perpendicular to the slope, crossing the blackberry and alder rows (Figure 2). Each transect held 12 sampling points in the alder and berry rows. To gain a better understanding of the spatial influence of the windbreak on topsoil characteristics, the distances between the samples close to the windbreaks were smaller (in every berry row), increasing towards the middle of the berry plots (serving as reference with no windbreak influence). The samples were taken in the topsoil (0-10 cm) with an ordinary hand shovel within the blackberry rows, in between individual plants, avoiding the direct root environment. One sample of about 300 g was taken per sampling point. The samples were homogenized, air-dried and sieved (<2 mm), and two sub-samples of each sample were transferred to Germany for laboratory analyses. Carbon and nitrogen contents were estimated at the Soil Ecology laboratory at the chair of Soil Ecology at the University of Freiburg, Germany, and soil texture was estimated at the laboratory of the research area Landscape Functioning at the Leibniz Centre for Agricultural Landscape Research (ZALF) in Müncheberg, Germany.

### 3.3.3   Analytical procedures

The analyses carried out on the samples differed between the samples depending on their research purpose. The profile samples were needed for soil classification while the transect samples were taken in order to study the spatial variability in soil characteristics.

We determined residual water content of all soil samples by drying an aliquot at 105 °C and comparing wet and dry weights of the sample as most analysis methods are related to dry mass of soil. The pH value of the samples was measured in a 1:2 soil-water slurry with ultra-pure water and with a glass electrode (Metrohm Titrino 751 GDP meter, Herisau, Switzerland).

To determine total carbon and nitrogen concentrations, subsamples of soil material were milled (Siebtechnik TEMA, Mülheim an der Ruhr, Germany), dried at 105 °C , combusted at 1150 °C and measured with an elemental analyzer (Vario EL cube, Elementar, Langeselbold, Germany). This was done for both the profile and the transect samples. Exchangeable base cations calcium (Ca), magnesium (Mg), potassium (K) and sodium (Na) and potential cation exchange capacity (CECpot) were extracted with 1 M $NH_4$-acetate at pH 7 according to the ammonium acetate method (van Reeuwijk, 2002) of the soil profile samples only. Cation concentrations in extracts were measured with an ICP-OE spectrometer (Spectro Ciros CCD ICP Side-on Plasma Optical Emission Spectrometer, Kleve, Germany). The percentage of base saturation was calculated from the results of exchangeable cations and CECpot.

The soil texture for the profile samples was determined after previous destruction of organic matter with combined method of sieving and sedimentation according to DIN ISO 11277 (ISO, 2002). This was carried out at the soil science laboratory, University of Freiburg (Germany).

The soil texture for the transect samples was analyzed without destroying organic material in order to obtain the effective particle size distribution along the transects throughout the cultivation system, and only the laser diffraction method was used. After using $Na_4P_2O_7$ as a dispersing agent, the sand fraction was obtained by sieving the soil solution through a 63 $\mu$m mesh. These sieved particles were dried at 105 °C , and the sand fraction was calculated as the ratio of sand to total dried soil sample

mass. A sub-sample of the suspension was transferred to the wet dispersion unit of the Mastersizer 3000 and stirred (3500 rpm and 100 % sonication power) until the laser obscuration level reached 10 %. Clay and silt contents were calculated according to the method for laser diffraction particle size analysis presented in Faé et al. (2019). This was carried out in the laboratory at ZALF, Müncheberg.

### 3.3.4 Data processing and quality control

Soil sampling was carried out according to the FAO Guidelines for soil description (Jahn et al., 2006). The analysis of physical and chemical soil parameters followed standard methods described in Procedures for Soil Analysis (van Reeuwijk, 2002) and according to DIN ISO 11277 (ISO, 2002). The quality of the nutrient analyses was continuously checked and calibrated using external standards. We used the Certipur, ICP multi-element standard solution IV (Merck KGaA, Darmstadt, Germany). Additionally we measured an internal soil standard with known nutrient contents together with the samples. The analyses of CECpot and exchangeable base cations were performed in triplicates.

### 3.4 Dendrological measurements

The research site was scanned in September 2019 with the RIEGL VZ 2000i terrestrial LiDAR (RIEGL Laser Measurement Systems GmbH, Horn, Austria), set with a pulse repetition rate of 600 kHz, and reported with a laser beam divergence of 0.27 mrad (an increase of 27 mm of beam diameter per 100 m distance). A multiple-scans approach was used under negligible wind conditions. We derived 3D point clouds from the scans and processed these to obtain structural tree data, foliage data, and windbreak characteristics. We also measured diameter at breast height (DBH) and between-tree spacing manually, in order to cross reference the point cloud data.

### 3.4.1 3D data processing

Co-registration was carried out using the software RiSCAN PRO 2.11.3 (RIEGL Laser Measurement Systems GmbH, Horn, Austria) and following the software guided-steps to coarse register point clouds in an outdoor non-urban scenario, while using multi-station Adjustment 2 for fine registration in the same software. In total, 42 scanning positions focusing on the windbreak structure were selected to represent the study area. These had a varied scanning step of 5 m to 30 m between scanning points. The single-scans points further than 60 m and isolated points were removed. The exported project point cloud was made homogeneous by applying cubic down-sampling (25 mm voxel side), and saved as a LAZ file (1.4 format) including many attributes of the laser data. A video animation is included with the data on the repository for an overview of the research area.

The windbreak point cloud included 32 scanning positions covering the target windbreak trees. From the linear windbreak structure, 14 scanning positions were within 10 m distance from the target windbreak, and distant from each other by 5 to 10 m. The remaining 18 positions were located between 15 and 25 m away from the windbreak base. Trees were partially under leaf-off conditions as scanning took place in early spring, but some trees had retained leaves from the previous season. The presence of obstacles within the site i.e. the support structures for the blackberry crop made movement within the field difficult

and formed obstructing structures within the laser's field of view, leading to occlusion of the deep crown and lower parts of the stem.

### 3.4.2 Tree data

The studied windbreak section was extracted from the co-registered project point cloud and filtered by pulse deviation (equal or lower than 10; Pfennigbauer and Ullrich, 2010)). Isolated points were removed and the windbreak point cloud was exported as a separate file. From within the windbreak point cloud, starting from the east-northeast edge in a west-southwest direction, 18 single-tree point clouds were manually segmented, thus isolating them from the point cloud as a whole, and sequentially labelled t01 to t018. These were further processed by filtering out foliage points (utilising reflectance and RGB values), while duplicated points, noisy and outlier points were removed using CloudCompare v2.10.2 (CloudCompare, 2019). Lastly, the individual tree point clouds were separated into three occlusion category classes regarding occlusion of the woody components, namely low, mid, and high occlusion levels.

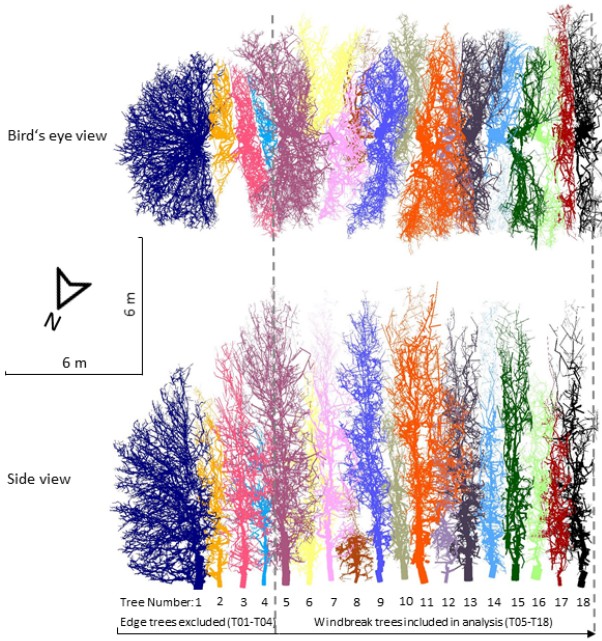

**Figure 7.** Bird's eye and side view of cylinder models of the sample windbreak. Colour denotes individual trees. Trees 1—4 (t01 to t04) were excluded from the analysis due to the windbreak edge influencing growth form.

The single-tree point clouds were used for producing cylinder models in the form of Quantitative Structure Models (QSMs) with the MATLAB implementation of TreeQSM v2.3 (Raumonen et al., 2013). Following the framework of Calders et al.

(2015), QSMs were optimised by testing 72 combinations of inputs over 25 runs, and best models were selected using the 'mean point-cylinder distance' as metric. Tested parameters were (in metres): PatchDiam1 (0.30, 0.25, 0.15, 0.10); PatchDiam2Min (0.015, 0.02, 0.03); PatchDiam2Max (0.05, 0.07, 0.09), and; lcyl (4, 6, non-metric). The optimised input parameters and the precision of the QSM-estimates are provided within occlusion category classes. Other QSM-parameters employed were FilRad (3.5) and nmin1 (4). Model output is shown in Figure 7, here side and bird's eye views of the windbreak are shown, individual trees are shown in different colours; note that trees t01 to t04 were not considered for analysis due to a differing growth form attributed to the edge effect. Volumetric tree parameters were converted to biomass using a wood density of 420 kg m$^{-3}$, considering an average value for *Alnus sp.* (after Harja et al., 2023). The coarse root biomass (roots with > 2 mm cross-sectional diameter) was estimated as 28.54 % of the total aboveground woody biomass (Frouz et al., 2015), and defined as belowground biomass, and also converted to volumetric terms.

### 3.4.3 Foliage data

Foliage data was retrieved by applying the 'leaf creation algorithm' (LCA) after Bohn Reckziegel et al. (2021) to the QSMs of trees t05 to t18. According to leaf sizes as described for *Alnus sp.* by European Atlas of Forest Tree Species (San-Miguel-Ayanz et al., 2016), the LCA was modified to restrict the use of the leaf classes to 'small', 'medium' and 'large' categories, with corrected proportions. The parameter leaf spacing was defined to values of 2.0, 2.5, 3.0 cm, and total leaf area was estimated on a tree basis. Leaf dry mass was estimated using the specific leaf mass of 13.3±0.3 m$^2$ kg$^{-1}$ for (*Alnus glutinosa* (L.) Gaertn.) as given by Johansson (1999).

### 3.4.4 Windbreak properties

Windbreak properties were assessed upon the acquired 3D data (terrestrial LiDAR), the assessment of tree structure with QSMs and the generated foliage data. A number of windbreak properties from the windbreak point cloud were derived from manipulation using CloudCompare v2.10.2 (CloudCompare, 2019), namely: length, width, tree number and tree spacing, for the entire windbreak. Average values for the QSM-derived attributes of measured trees were also computed, for the trees t05 to t18. Tree volume and biomass, as well as total leaf area and leaf mass, were estimated per metre of windbreak. Plant coverage was derived as the ratio of the alpha-convex hull (ahull function from the 'alphahull' package by Pateiro-Lopez et al. (2019)) of leaf points, and the minimum binding box (getMinBBox function from the 'shotGroups' package by Wollschlaeger (2022)) in R v4.0.4 (R Core Team, 2021). Leaf area index (LAI) was derived as the total leaf area by the canopy projection area, defined as an alpha-convex hull.

### 3.4.5 Data processing and quality control

The processing of the 3D data followed a standard protocol within terrestrial LiDAR analyses (e.g. Raumonen et al., 2013). QSM precision was assessed utilising occlusion categories based on input parameters within the QSM modelling process as described above. Due to the highly dense vegetation and field obstacles, a complete coverage of the windbreak was impeded.

Optimisation was applied to control the precision of the tree reconstruction. The validation of tree measurements would require destructive sampling of trees which was not permitted, nevertheless past studies have shown high correlation between traditional measurement techniques and LiDAR derived data (e.g. Kuekenbrink et al., 2021; Calders et al., 2015; Raumonen et al., 2015). The non-destructive foliage data simulation used the LCA (Bohn Reckziegel et al., 2021) and was facilitated by a robust reconstruction of the trees with optimised QSMs. Nevertheless, the LCA results may be subject to errors, i.e. at the top of the QSMs due to lower point density, and due to varying foliage capacity of a tree (ecophysiological state). The description of tree windbreaks with parameters derived from 3D data assist the characterisation of these vegetative structures, enabling the estimation of many hard-to-measure tree attributes.

## 4   Applications

The multidisciplinary focus of the complete dataset package allows for various analysis avenues. The main research goal we wanted to target by collecting this data was to assess the influence of the windbreak in this AFS onto water fluxes, nutrient distribution and microclimate in the crop. The corresponding scientific study is Hoffmeister et al. (2023). By combining the different measurements the effects of the windbreak were clearly visible, as well as the effect of the irrigation in the system. The latter influenced the water balance by reducing water limitation, securing sufficient water for plant growth. The windbreak also needed water to grow, however, it also lowered the irrigation demand for the blackberry crop by reducing soil evaporation and influencing water redistribution. Nutrient accumulation near the windbreak and occurrence of water erosion were indicated by soil physiological properties and nutrient distribution. Additionally, the windbreak increased the carbon sequestration potential compared to monoculture farming.

The detailed information on site conditions as well as the high-resolution time series of meteorological parameters, soil moisture and matric potential enable hydrological modelling studies on the slope scale. Once parameterised, scenarios can be run to elucidate the shifts between water limitation and energy limitation in the AFS and to inform about possible management strategies to utilise the windbreak influence (Hoffmeister et al., in prep.).

The availability of the point cloud data acquired with terrestrial LiDAR facilitates the three-dimensional examination of the studied windbreak in high detail. This allows for the scrutiny of sections of the windbreak row, individual trees or tree parts thereof, and can be carried out either within the context of the study of AFS, *Alnus sp.* or for those interested in vegetative windbreak structures. Moreover, the data set provides a unique view on a frequent, though not often understood, and alternative spatial arrangement of trees with specific tree growth conditions, which differ from forest formations, stands and scattered trees. The point cloud data set offers the possibility to use it as an independent training data set limited in size and spatial complexity.

The variety of measurements across different ecosystem compartments also supports studying the interactions between these compartments which is rarely possible in otherwise mostly discipline-specific campaigns. Additionally, the measurements serve as an example of a (for the region typical) AFS including windbreak and crop and and can be compared to other windbreaks and to other AFS systems. This enables further insights into the varied interactions between the different components of these systems to help elucidate benefits and limitations of introducing AFS as a sustainable landuse option in a changing climate.

Limitations of the dataset include its limited geographical scope, as we only measured at one AFS on one slope at the one location. Further specifics of the site should also be kept in mind when using the data. For example, the irrigation scheme, of which we only have estimates and no accurate values, will affect any water attepted balance calculations. Additionally, the meteorological variables are affected by the windbreak influence on the wind field.

## 5 Data availability

The described datasets are freely available and accompanied by a technical description of the individual tables from the online repository GFZ Data Services and are published under https://dataservices.gfz-potsdam.de/panmetaworks/review/eb7c7f 03ca6d02f7d650884baf4633a8f94eaf4b1c2dfb025f1671c5dcba2638 (Hassler et al., 2024). The data files in the repository are named according to their respective research discipline meteorology ("_Meteo_"), hydrology ("_Hydro_"), soil science ("_Soil_"), dendrology ("_Dendro_"). They include time series for the meteorological variables, soil moisture and matric potential as well as data from the joint measurement campaign, namely nutrient, cation and texture data for the soil samples, soil hydraulic characteristics from the cylinder samples and the derived tree and windbreak characteristics from the terrestrial laser scans.

## 6 Conclusions

Interdisciplinary and multivariate field campaigns are notoriously challenging, expectations and requirements for the field sites vary between the different groups and there are never enough measurements that can be taken to really cover all aspects of the ecosystem satisfactorily. Our dataset was intended as an example to enable studying various influences of the tree component in this AFS, and also to be compared to similar measurements in other AFS. Despite some limitations, it provides a glimpse in the relevant processes and opens up further avenues to assess and promote AFS as a sustainable land use type in a changing world.

 **Appendix A: Overview of meteorological time series**

We provide an overview of the time series of meteorological measurements (A1), primarily to give a first idea on what kind of data, their dynamics and ranges are included in the data files. Some of most likely less important time series (maximum precipitation rate, temperature of the humidity sensor, wind direction, gust speed and lightning count and distance) are omitted from the figure to make it clearer.

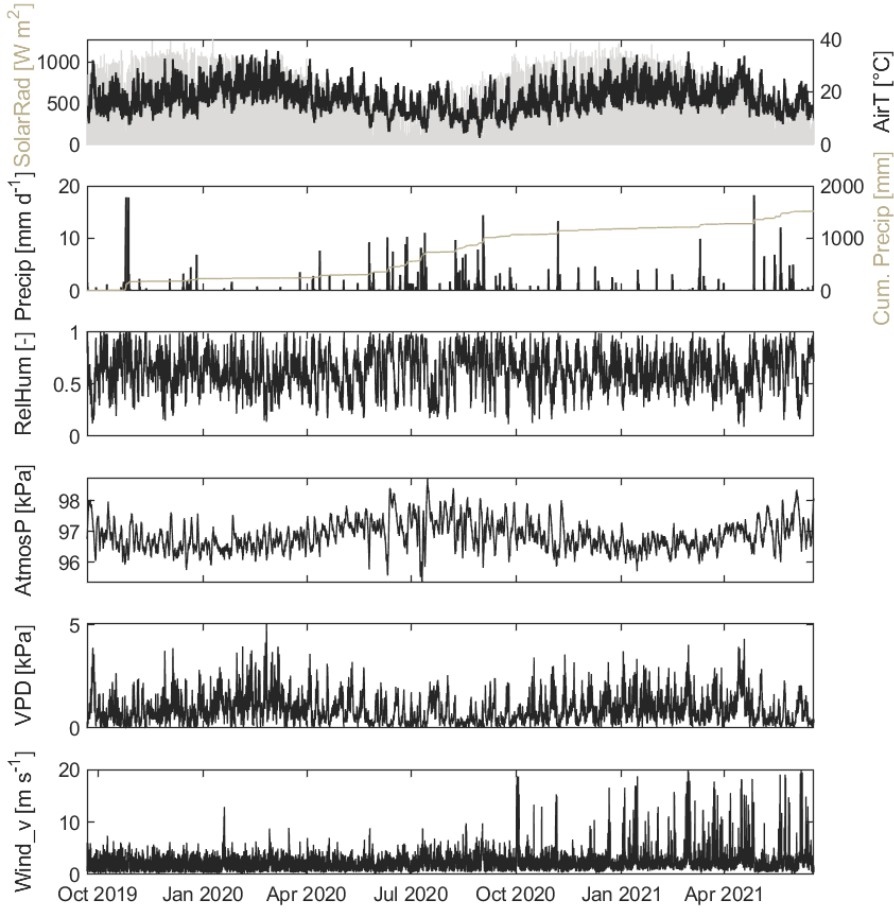

**Figure A1.** Time series of meteorological variables over the course of the monitoring period. Shown are Solar radiation (SolarRad), air temperature (AirT), precipitation (Precip), cumulative precipitation (Cum. Precip), relative humidity (RelHum), atmospheric pressure (AtmosP), vapour pressure deficit (VPD) and wind velocity (Wind _v).

*Author contributions.* SKH, RBR, SH, FK, RM and JPS conceived, planned and carried out the data collection. BdT and AK provided local support and assistance with measurements and maintenance. All authors provided critical feedback and helped shape the presentation of data and manuscript.

*Competing interests.* The first author is member of the editorial board of Earth System Science Data.

*Acknowledgements.* The collection of this dataset would not have been possible without the support of Raymond O'Grady and staff at Hillcrest Berries (Pty) Ltd who permitted access and accommodated the installation of equipment and long-term measurement within a working and productive farm environment. We are very appreciative of the support. We also want to acknowledge the contribution of our colleagues at the Department of Forest and Wood Sciences, Stellenbosch University, namely Deon Malherbe and Cláudio Cuaranhua who provided invaluable knowledge of the local site conditions, logistics support and equipment maintenance as well as data download. The research was funded by the German Federal Ministry of Education and Research (BMBF) with the grant number 01LL1803 as part of the project Agroforestry in Southern Africa: new pathways of innovative land use systems under a changing climate (ASAP). The article processing charges for this publication were covered by the open access publishing fund of the Karlsruhe Institute of Technology (KIT) in the Helmholtz Association.

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
