# Peer review of "Multivariate characterisation of a blackberry-alder agroforestry system in South Africa: Hydrological, pedological, dendrological and meteorological measurements"

_Earth System Science Data, 2023_

## Referee Comment (RC1)

**Review to Hassler et al.: "Multivariate characterisation of a blackberry-alder agroforestry system in South Africa: Hydrological, pedological, dendrological and meteorological measurements"**

**General comments:**
In the paper "Multivariate characterisation of a blackberry-alder agroforestry system in South Africa: Hydrological, pedological, dendrological and meteorological measurements" an impressive data set is presented. The data set was collected at a unique agroforestry site, characterised by a diverse orientation of tree strips and fairly complex terrain. The authors made a great effort to get a big picture of the site and its hydrological, pedological, dendrological and meteorological conditions. Overall the paper is well written, is concise and kept short. But, I found some inaccuracies in the text and in the data itself, which I think have to be clarified. Please correct these before the paper can be published and the data can easily be used by users. I wish the authors best of success with the publication of the paper.

**Specific comments:**
1. Introduction: For me the linkage between the experiment in South Africa and agroforestry is missing. I think it would be good to mention which environmental problems occur at that region and how agroforestry can solve and mitigate these problems. I see that you, in a more general way, mentioned/ listed positive effects of AF, but as I said, it would be good to be more specific from the beginning.

2. All sections on data processing and quality control: I am missing a more detailled description on how e.g. the meteorolgical and soil data were filtered, using e.g. upper and lower limits (and which) or other despiking routines, such that the user knows about the quality of the data. Here you mainly report on wind speed and relatively vague on other parameter, but often not specific enough.

3. I am aware that the data are linked to the paper of Hoffmeister et al. (2023) and that some data are presented there. From my perspective it would be nice to include a results section, where you present timeseries of the relevant parameter mentioned, such that I as an intended data user get an idea on how the data look like and about their quality. This would later on also support the "Application" section, where you can link your data/figure and the suggested future applications.

4. I also had a look on the data and found some errors. In parallel I checked the document "2023-028_Hassler-et-al_Data_Description.pdf", which is not always in line with the data found in the files. I am wondering, whether the published data are raw data or whether these are filtered? Otherwise I suggest checking the data again and updating the description file. From a user perspective this would be of high value.

Details on the files are:

*2023-028_Hassler-et-al_Meteo_MeteorologicalMeasurements.csv:*
i) Wind velocity (Wind_v) is not in m s$^{-1}$, the magnitude is off and especially at the end of 2020 and whole 2021 there are many spikes, which might correspond to the filtered data for wind speed > 30 m s$^{-1}$
ii) Gust_v: values of 30 m s$^{-1}$ at the end of 2020 and whole 2021 are not realistic and seems to be not filtered

iii) MaxPrecipRate: In the document it says mm h⁻¹ and in the paper it says "maximum rate for each 10-minute interval." So what is the truth?

*2023-028_Hassler-et-al_Hydro_SoilMoistureMatricPotential.csv:*
i) the soil water content is in % and not in $m^3\ m^{-3}$
ii) spikes in ThetaWB60, ThetaWB80, ThetaWB10, ThetaB20, ThetaB40, ThetaB60, ThetaB80, TmatPotWB10, MatpotWB30, MatpotWB40, TmatpotWB40 should be removed

iii) I am wondering why the matric potential is always positive, shouldn't it be negative, as you also state in the paper at l. 175?

**Technical corrections:**
l. 12: The sentence sound weird: "The provided data is intended to explore the interaction between trees and crops in agroforestry landscapes."

*Maybe better:*
The provided dataset can be used to explore the interaction between trees and crops in agroforestry landscapes.

And I would also add an additional sentence as outlook on the potential use of the data set for different user, similar to what you've done in section 4 'Applications', but just summarised.

l. 29: please include a reference after this sentence: "By applying...it is demonstrated that AFS …. on an equivalent land area (e.g. … citation)."

l. 30: same here, include reference please

l. 40: separate sentences: … (Hombegowda et al., 2016). The presence…

l. 44: The question is, which temperature and humidity you refer to; I guess air temperature and humidity, if so, please write this, e.g.: … both *air* temperature and *air* humidty…

l. 47: plural: … in agricultural *fields*

l. 76: … focused *on* two …

Caption Figure 1: rewrite caption: Images from the field site taken in a) winter and b) late spring.

l. 85: remove second *and*

l. 86: which manual measurements of the windbreak trees were done? Have you measured the BHD or what else? Clarify please.

l. 91: inlcude city and country of company: (METER Group,?,?) and also here ZL6 cloud Data Logger (company?, city?, country?)

l. 93-108: I suggest to place the list of variables into a table. This would be easier to follow. You could then include the range, the resolution, the accuracy and the unit each in one column. And also include all other inctruments described in the text and then include the company, city and country. This would be consistent.

l. 94: is 1750 W m⁻² the maximum possible value the sensor can measure or is this also the value you used to filter the data for outlier? If the latter applies keep in mind that global radiation can be max 1360 W m⁻².

Caption Figure 2: a point is missing at the end.

l. 98: what are the values in brackets behind 0-100% (0-1.0), indicate! How is the accuracy changing with temperature and humidity? Please indicate!

l. 100: out of curiosity: what is the temperature of the humidity sensor? Is there a second temperature sensor, e.g. used to calculate VPD? Or is this rather the dewpoint temperature calculated out of RH?

l. 122: comma missing: Additionally, …

l. 124:
→ as mentioned already, maybe include the sensor information into one common table.
→ Here also be consistent in how you separate sensor name and company, use a comma instead of a semicolon.
→ Also rewrite this sentence, e.g.: Soil moisture was monitored continuously … accuracy: +- 2%) in two profiles from … . If it is a profile, maybe direcly mention the different installation depths in brackets behind.
→ from my point of view TDR is also an abbreviation, which you also used in Fig. 3, so please once write TDR out completely and abbreviate in brackets behind.

l. 128: include comma: In each tube, …

l. 149-152: I guess values in mm and µm refer to particle diameter, write this maybe like $d_p$=…. With $d_p$ the particle diameter

l. 153: include country (UMS GmbH, Munich, Germany).

Figure 4: Write θ instead of Theta, this is more consistent with Eq. (1).

l. 165: please include units for the different water contents or state that the soil water content is in either m³ m⁻³ or %.

l. 173: what does this standard soil calibration refer to? Is the soil density of 1.40675 g cm⁻³ used in an equation to convert from e.g. raw voltage/resistance into soil water content? If an equation is used I suggest including it here.

Figure 5: → … according to *the* IUSS …
→ reference to sub-figures a) and b) is missing in the caption

l. 202 ff: split sentence in two and rewrite the sentence, this sounds weird. One suggestion would be:

The samples were homogenized, air-dried and sieved (<2 mm), and two sub-samples of each sample were transferred to Germany for laboratory analyses. Carbon and nitrogen contents *were estimated at* the Soil Ecology laboratory at the chair of Soil Ecology at the University of Freiburg, Germany, and soil texture was *estimated* at the laboratory of the research area Landscape Functioning at the  Leibniz Centre for Agricultural Landscape Research (ZALF) in Müncheberg, Germany.

l. 213: include city and country of company (Siebtechnik...)

l. 214: include comma: … dried at 105°C, combusted …

l. 220: delete "two methods"

l. 222: … University of Freiburg, Germany.

l. 226: These sieved particles were dried at 105°C [space] and the *sand fraction* [check missing text] was calculated …. .

l. 229: … particle size analysis *presented* in …

l. 233: remove space between bracket and point.

l. 234: which external standards were used?

l. 236: include city and country of company behind LiDAR (city, country).

l. 241: include reference to software and what is the standard protocol?

l. 245: specify where this video is included? E.g.: A video animation is included to the data set for …

l. 255: Split and rewrite sentence: The *studied* windbreak section was extracted from the co-registered project point cloud *and* filtered by pulse deviation (equal or lower than 10; Pfennigbauer and Ullrich, 2010). The isolated points were removed and exported as a separate file.

l. 266-267: What do the values in brackets refer to? What is the unit? Diameter maybe m?

l. 272: (>2mm) what do this refer to? Is it root diameter? If yes, please indicate.

Figure 6: include indication on position of sub-figure in caption, e.g.: Bird's eye *(top)* and side view *(bottom)* of …

l. 288 and l. 289: include reference behind the R-packages

l. 306: it is not crops your are referring to, aren't these shrubs or better just berry bushes? I think "… nutrient distribution and microclimate on the berry bushes." is better suited.

l. 306-312:

→ Here you seem to refer to results from Hoffmeister et al. (2023). Present these results in past tense.

→ l. 306: By combining… This paragraph sounds out of place without a figure. From my perspective it doesn't make sense without presenting the results or at least some time series. Maybe it would be wise to include a results section in addition to/ before the Applications section?! This would also be different from results presented in Hoffmeister et al. (2023) and would give a quick overview on how the data look like. See also my comment at the beginning.

l. 317: TLS not defined: write out and abbreviate and "scans" can be deleted, as this is included in TLS

---

## Author Comment (AC2)

**Response to review 1 to Hassler et al.: "Multivariate characterisation of a blackberry-alder agroforestry system in South Africa: Hydrological, pedological, dendrological and meteorological measurements"**

Original review comments in black

*Answers in blue italics*

General comments:

In the paper "Multivariate characterisation of a blackberry-alder agroforestry system in South Africa: Hydrological, pedological, dendrological and meteorological measurements" an impressive data set is presented. The data set was collected at a unique agroforestry site, characterised by a diverse orientation of tree strips and fairly complex terrain. The authors made a great effort to get a big picture of the site and its hydrological, pedological, dendrological and meteorological conditions. Overall the paper is well written, is concise and kept short. But, I found some inaccuracies in the text and in the data itself, which I think have to be clarified. Please correct these before the paper can be published and the data can easily be used by users. I wish the authors best of success with the publication of the paper.

*Thank you very much for the detailed review of paper, data and data description from a user perspective and for acknowledging the value of the dataset! We address your comments in the following and are confident the revisions based on your suggestions will help to ensure an easier and broader use of the data.*

**Specific comments:**

1. Introduction: For me the linkage between the experiment in South Africa and agroforestry is missing. I think it would be good to mention which environmental problems occur at that region and how agroforestry can solve and mitigate these problems. I see that you, in a more general way, mentioned/ listed positive effects of AF, but as I said, it would be good to be more specific from the beginning.

*We will make the following additions and changes with citations to support them:*

*Line 24: ……..drought and extreme climatic events are expected to become more frequent (New et al., 2006; Meadows, 2006; IPCC, 2014). South Africa particularly, is expected to experience shifting rainfall patterns and increased temperatures, therefore, climate change adaptation strategies, the implementation of AFS being one particular tool in the toolbox, are of the highest importance to mitigate future pressures (Veste et al, 2024; Ziervogel et al. 2014).*

*Line 27: ……….human needs with a more natural resource management approach which is considered important, particularly in South Africa (Sheppard et al., 2020, Zerihun, 2021).*

2. All sections on data processing and quality control: I am missing a more detailed description on how e.g. the meteorological and soil data were filtered, using e.g. upper and lower limits (and which) or other despiking routines, such that the user knows about the quality of the data. Here you mainly report on wind speed and relatively vague on other parameter, but often not specific enough.

*For the most part the reported data is raw data where we only removed very obvious erroneous values that are attributable to sensor/logger failure etc. For the variables where we did pre-*

*processing, we will explicitly state so in the data processing descriptions and explain the details of the processing .*

3. I am aware that the data are linked to the paper of Hoffmeister et al. (2023) and that some data are presented there. From my perspective it would be nice to include a results section, where you present timeseries of the relevant parameter mentioned, such that I as an intended data user get an idea on how the data look like and about their quality. This would later on also support the "Application" section, where you can link your data/figure and the suggested future applications.

*Thank you for the suggestion. We agree that on overview of the data will help the readers to assess the possible applications and might increase usage of the data. We will include figures of the time series of the meteorological and hydrological variables. We will not put them in a separate "Results" section though, firstly because it's quite uncommon in a data paper and secondly because it's more a visualization of (almost raw) data and not a result of any analysis. We will put the figures with the respective sections that describe the measurements.*

4. I also had a look on the data and found some errors. In parallel I checked the document "2023-028_Hassler-et-al_Data_Description.pdf", which is not always in line with the data found in the files. I am wondering, whether the published data are raw data or whether these are filtered? Otherwise I suggest checking the data again and updating the description file. From a user perspective this would be of high value.

*Thank you very much for also checking the data files and the consistency between description, data and manuscript. We will also re-check the files to make sure they properly correspond to each other. We respond to your detailed comments below.*

**Details on the files are:**

*2023-028_Hassler-et-al_Meteo_MeteorologicalMeasurements.csv:*

i)      Wind velocity (Wind_v) is not in m s $^{-1}$, the magnitude is off and especially at the end of 2020 and whole 2021 there are many spikes, which might correspond to the filtered data for wind speed > 30 m s$^{-1}$
        *The values are indeed wind velocity in m s$^{-1}$, however, as we state in the data processing section, the sensor can have issues with moisture and then shows unrealistic values. We will go through the data once more to find and remove all of them.*

ii)     Gust_v: values of 30 m s$^{-1}$ at the end of 2020 and whole 2021 are not realistic and seems to be not filtered
        *We will also go through the gust data one more time and remove any questionable values. However, gusts with high wind speed were also recorded in other reference stations around the site. We will these stations again for consistency.*

iii)    MaxPrecipRate: In the document it says mm h $^{-1}$ and in the paper it says "maximum rate for each 10-minute interval." So what is the truth?
        *Thank you for pointing this out. That is an error on our site. Precipitation is given as "accumulation over number of minutes (interval) 1-min periods" and MaxPrecipRate is the "accumulation recorded for a single 1-min period multiplied by 60 min/h".*
        *We will change the text in the paper accordingly.*

2023-028_Hassler-et-al_Hydro_SoilMoistureMatricPotential.csv:

i)      the soil water content is in % and not in m³ m$^{-3}$
        *Thank you for noticing this. We will correct the units in the revised manuscript.*

ii) spikes in ThetaWB60, ThetaWB80, ThetaWB10, ThetaB20, ThetaB40, ThetaB60, ThetaB80, TmatPotWB10, MatpotWB30, MatpotWB40, TmatpotWB40 should be removed

*We will remove the spikes of the obvious erroneous measurements and update the data and their description respectively. However, we would not do extensive data pre-processing, we only remove the values that we can definitely attribute to sensor/logger faults etc. All other values that are plausible would remain in the data.*

iii) I am wondering why the matric potential is always positive, shouldn't it be negative, as you also state in the paper at l. 175?

*Yes. The numbers are positive in the data file (which is a common way to report them), but matric potential is always negative. We will clarify this in the revised version.*

**Technical corrections:**

l. 12: The sentence sound weird: "The provided data is intended to explore the interaction between trees and crops in agroforestry landscapes."

*Maybe better:*

The provided dataset can be used to explore the interaction between trees and crops in agroforestry landscapes.

And I would also add an additional sentence as outlook on the potential use of the data set for different user, similar to what you've done in section 4 'Applications', but just summarised.

*Thank you for the suggestion, we will change the text as follows:*

*"The dataset provides a multidisciplinary approach to assess the impact and interaction of windbreaks and tree structures in agroforestry landscapes, aiding future work concerning water fluxes, nutrient distribution, microclimate, and carbon sequestration. The dataset, including high-resolution time series and point cloud data, offers valuable insights for managing the windbreak's influence and serves as a unique training dataset for spatial analysis."*

*We will also add some sentences on the potential limitation of the dataset, e.g. due to the influence of irrigation, to help the users to assess the potential applications better.*

l. 29: please include a reference after this sentence: "By applying...it is demonstrated that AFS …. on an equivalent land area (e.g. … citation)."

*We will modify the sentence before according to your previous comment (Specific comment #1), and put the sentence on the land equivalent ratio with the respective citation (Ong and Kho, 2015) in a separate sentence. This should clarify this passage.*

l. 30: same here, include reference please

*This sentence still refers to the LER concept by Ong and Kho (2015). However, as the citation is added already in the previous sentence we would consider the concept sufficiently cited and not cite the same paper again in this sentence.*

l. 40: separate sentences: … (Hombegowda et al., 2016). The presence…

*We will do this in the revision.*

l. 44: The question is, which temperature and humidity you refer to; I guess air temperature and humidity, if so, please write this, e.g.: … both air temperature and *air* humidty…

*We will correct this in the revision.*

l. 47: plural: … in agricultural *fields*

*We will correct this in the revision.*

l. 76: … focused *on* two …

*We will correct this in the revision.*

Caption Figure 1: rewrite caption: Images from the field site taken in a) winter and b) late spring.

*We will change this in the revision.*

l. 85: remove second *and*

*We will correct this in the revision.*

l. 86: which manual measurements of the windbreak trees were done? Have you measured the BHD or what else? Clarify please.

*We measured diameter at breast height (DBH) and between-tree spacing in order to cross reference the point cloud data. We will add this information in the revision, in the tree measurement section 3.4.*

l. 91: include city and country of company: (METER Group,?,?) and also here ZL6 cloud Data Logger (company?, city?, country?)

*We will add this in the revision.*

l. 93-108: I suggest to place the list of variables into a table. This would be easier to follow. You could then include the range, the resolution, the accuracy and the unit each in one column. And also include all other instruments described in the text and then include the company, city and country. This would be consistent.

*Thank you for the suggestions. We will put the measured parameters, instruments, companies, range, accuracy and resolution in a table, one table for the meteorological measurements and one for the hydrological time series. For the laboratory analyses it does not really seem to be useful. The units of the measurements are reported as part of the detailed description of the data tables in the "Data description" file at the repository.*

l. 94: is 1750 W m $^{-2}$ the maximum possible value the sensor can measure or is this also the value you used to filter the data for outlier? If the latter applies keep in mind that global radiation can be max 1360 W m$^{-2}$.

*The value of 1750 W m$^{-2}$ is indeed the maximum possible value the sensor can measure which, as you rightly pointed out, is quite high given that the solar constant is 1361 W m$^{-2}$. We used the solar constant as a reference to filter the data for outliers, but fortunately all the values seem reasonable.*

Caption Figure 2: a point is missing at the end.

*We will add it in the revision.*

l. 98: what are the values in brackets behind 0-100% (0-1.0), indicate! How is the accuracy changing with temperature and humidity? Please indicate!

*The values in brackets refer to the percentage, which, as usual, is given as a value between 0 and 1 in the CSV file.*

*The accuracy of the measurement is slightly lower when the relative humidity is above 80%. We will add a short explanation to the manuscript to clarify this.*

l. 100: out of curiosity: what is the temperature of the humidity sensor? Is there a second temperature sensor, e.g. used to calculate VPD? Or is this rather the dewpoint temperature calculated out of RH?

*Yes, there is a second temperature sensor. The vapour pressure sensor is located behind a circular Teflon screen in the same housing as the sonic transducers and measures relative humidity and temperature in addition to computing vapour pressure.*

l. 122: comma missing: Additionally, …

*We will add it in the revision.*

l. 124:

→ as mentioned already, maybe include the sensor information into one common table.

*Yes, we will put sensor information in tables, however, separately for the meteorological and hydrological time series.*

→ Here also be consistent in how you separate sensor name and company, use a comma instead of a semicolon.

*Thank you for noticing. We will check for consistency in the sensor/company references in the revision.*

→ Also rewrite this sentence, e.g.: Soil moisture was monitored continuously … accuracy: +- 2%) in two profiles from … . If it is a profile, maybe directly mention the different installation depths in brackets behind.

*We will adapt the first sentence in the revision. As the "profile" refers to the profile probe in the acrylic tubes which we describe in the following sentences, we agree it might be confusing. We will change the text to "monitored continuously… at two locations in four different depths, from…".*

→ from my point of view TDR is also an abbreviation, which you also used in Fig. 3, so please once write TDR out completely and abbreviate in brackets behind.

*We will add the full name for the time domain reflectometry in the revision.*

l. 128: include comma: In each tube, …

*We will add this in the revision.*

l. 149-152: I guess values in mm and µm refer to particle diameter, write this maybe like dp=…. With dp the particle diameter

*Thank you for the comment. We will add the term particle diameter (dp) as you suggest to make it clearer.*

l. 153: include country (UMS GmbH, Munich, Germany).

*We will add this in the revision.*

Figure 4: Write θ instead of Theta, this is more consistent with Eq. (1).

*We will adapt this in the revision.*

l. 165: please include units for the different water contents or state that the soil water content is in either $m^3\ m^{-3}$ or %.

*Thank you for noticing this. We will harmonize description and data units in the revision.*

l. 173: what does this standard soil calibration refer to? Is the soil density of $1.40675\ g\ cm^{-3}$ used in an equation to convert from e.g. raw voltage/resistance into soil water content? If an equation is used I suggest including it here.

*The devices' output is directly in vol. WC [%]. The conversion from electrical raw values to soil moisture are done internally, assuming by default a soil density of $1.40675\ g\ cm^{-3}$. For details, readers are referred to the manufacturer's manual, which also offers suggestions to account for density deviations.*

Figure 5: → … according to *the* IUSS …

*We will add this in the revision.*

→ reference to sub-figures a) and b) is missing in the caption

*We will add the details for the sub-figures in the revision: a) Profile with the delineated horizons, b) profile pit in between the berry rows.*

l. 202 ff: split sentence in two and rewrite the sentence, this sounds weird. One suggestion would be:

The samples were homogenized, air-dried and sieved (<2 mm), and two sub-samples of each sample were transferred to Germany for laboratory analyses. Carbon and nitrogen contents *were estimated* at the Soil Ecology laboratory at the chair of Soil Ecology at the University of Freiburg, Germany, and soil texture was *estimated* at the laboratory of the research area Landscape Functioning at the Leibniz Centre for Agricultural Landscape Research (ZALF) in Müncheberg, Germany.

*Thank you for the suggestion, we will change these sentences to your version in the revision.*

l. 213: include city and country of company (Siebtechnik…)

*It's Siebtechnik TEMA, Mülheim an der Ruhr, Germany. We will add it in the revision.*

l. 214: include comma: … dried at 105°C, combusted …

*We will add it in the revision.*

l. 220: delete "two methods"

*We will correct this in the revision.*

l. 222: … University of Freiburg, Germany.

*We will include the location as suggested (Germany).*

l. 226: These sieved particles were dried at 105°C [space] and the *sand fraction* [check missing text] was calculated …. .

*Thank you. We will correct this in the revision.*

l. 229: … particle size analysis *presented* in …

*We will correct this in the revision.*

l. 233: remove space between bracket and point.

*We will correct this in the revision.*

l. 234: which external standards were used?

*We used the Certipur, ICP multi-element standard solution IV (Merck KGaA, Darmstadt, Germany). Additionally we measured an internal soil standard with known nutrient contents together with the samples.*

*We will add this information to the manuscript.*

l. 236: include city and country of company behind LiDAR (city, country).

*We will add this information: (RIEGL Laser Measurement Systems GmbH, Horn, Austria)*

l. 241: include reference to software and what is the standard protocol?

*We will add the software information which is by the same company as the instrument: (RIEGL Laser Measurement Systems GmbH, Horn, Austria). Also, we will exchange the "standard protocol" with the more detailed explanation: ... and following the software guided-steps to coarse register point clouds in an outdoor non-urban scenario, while using multi-station Adjustment 2 for fine registration in the same software.*

l. 245: specify where this video is included? E.g.: A video animation is included to the data set for …

*We will add "…is included with the data on the repository for…" in the revision.*

l. 255: Split and rewrite sentence: The *studied* windbreak section was extracted from the co-registered project point cloud *and* filtered by pulse deviation (equal or lower than 10; Pfennigbauer and Ullrich, 2010). The isolated points were removed and exported as a separate file.

*Thank you for the suggestion, we will include your text improvement, slightly adapting the last sentence: "Isolated points were removed and the windbreak point cloud was exported as a separate file."*

l. 266-267: What do the values in brackets refer to? What is the unit? Diameter maybe m?

*Yes, the values are in metres. We will add this directly after "Tested parameters were (in metres).*

l. 272: (>2mm) what do this refer to? Is it root diameter? If yes, please indicate.

*This refers to cross-sectional diameter. We will adapt this to "(roots with > 2 mm cross-sectional diameter)" in the revision.*

Figure 6: include indication on position of sub-figure in caption, e.g.: Bird's eye *(top)* and side view *(bottom)* of …

*We will change this as suggested to:*

*"Figure 6. Bird's eye (top) and side view (bottom) of cylinder models of the sample windbreak. Colour denotes individual trees. Trees 1—4 (t01 to t04) were excluded from the analysis due to the windbreak edge influencing growth form."*

l. 288 and l. 289: include reference behind the R-packages

*We will add these in the revision.*

l. 306: it is not crops your are referring to, aren't these shrubs or better just berry bushes? I think "…nutrient distribution and microclimate on the berry bushes." is better suited.

*We agree that it is not a "typical" annual crop but a perennial one. However, the term is wide and can even be applied to trees in forests or for biomass. We therefore leave this unchanged.*

l. 306-312:

→ Here you seem to refer to results from Hoffmeister et al. (2023). Present these results in past tense.

*We will correct this in the revision.*

→ l. 306: By combining… This paragraph sounds out of place without a figure. From my perspective it doesn't make sense without presenting the results or at least some time series. Maybe it would be wise to include a results section in addition to/ before the Applications section?! This would also be different from results presented in Hoffmeister et al. (2023) and would give a quick overview on how the data look like. See also my comment at the beginning.

*We will add overview graphs of the time series so that the reader can quickly grasp what the data looks like. However as it is not strictly a result from an analysis and rather visualization of data, we would refrain from introducing a results section. We will most likely add time series graphs for important meteorological variables and the ones for soil moisture/matric potential to the respective sections where the data are described.*

l. 317: TLS not defined: write out and abbreviate and "scans" can be deleted, as this is included in TLS

*We will replace this by "The availability of the point cloud data acquired with terrestrial LiDAR facilitates the three-dimensional examination of the studied windbreak in high detail…"*

References:

Veste, M., Sheppard, J. P., Abdulai, I., Ayisi, K. K., Borrass, L., Chirwa, P. W., Funk, R., Kapinga, K., Morhart, C., Mwale, S. E., Ndlovu, N. P., Nyamadzaw, G., Nyoka, B. I., Sebola, P., Seifert, T., Senyolo, M. P., Sileshi, G. W., Syampungani, S., and Kahle, H.-P.: The Need for Sustainable Agricultural Land-Use Systems: Benefits from Integrated Agroforestry Systems, pp. 587–623, Springer International Publishing, Cham, https://doi.org/10.1007/978-3-031-10948-5_21 , 2024

Zerihun, M. F.: Agroforestry Practices in Livelihood Improvement in the Eastern Cape Province of South Africa, Sustainability, 13, https://doi.org/10.3390/su13158477 , 2021.

Ziervogel, G., New, M., Archer van Garderen, E., Midgley, G., Taylor, A., Hamann, R., Stuart-Hill, S., Myers, J., and Warburton, M.: Climate change impacts and adaptation in South Africa, WIREs Climate Change, 5, 605–620, https://doi.org/10.1002/wcc.295, 2014

---

## Author Response (AR1)

**Response to review 1 to Hassler et al.: "Multivariate characterisation of a blackberry-alder agroforestry system in South Africa: Hydrological, pedological, dendrological and meteorological measurements"**

Original review comments in black

*Answers in blue italics*

General comments:

In the paper "Multivariate characterisation of a blackberry-alder agroforestry system in South Africa: Hydrological, pedological, dendrological and meteorological measurements" an impressive data set is presented. The data set was collected at a unique agroforestry site, characterised by a diverse orientation of tree strips and fairly complex terrain. The authors made a great effort to get a big picture of the site and its hydrological, pedological, dendrological and meteorological conditions. Overall the paper is well written, is concise and kept short. But, I found some inaccuracies in the text and in the data itself, which I think have to be clarified. Please correct these before the paper can be published and the data can easily be used by users. I wish the authors best of success with the publication of the paper.

*Thank you very much for the detailed review of paper, data and data description from a user perspective and for acknowledging the value of the dataset! We addressed your comments in the following and are confident that the revisions based on your suggestions will help to ensure an easier and broader use of the data.*

**Specific comments:**

1. Introduction: For me the linkage between the experiment in South Africa and agroforestry is missing. I think it would be good to mention which environmental problems occur at that region and how agroforestry can solve and mitigate these problems. I see that you, in a more general way, mentioned/ listed positive effects of AF, but as I said, it would be good to be more specific from the beginning.

*We made the following additions with citations to support them:*

*Line 24: ........drought and extreme climatic events are expected to become more frequent (New et al., 2006; Meadows, 2006; IPCC, 2014). South Africa particularly, is expected to experience shifting rainfall patterns and increased temperatures, therefore, climate change adaptation strategies, the implementation of AFS being one particular tool in the toolbox, are of the highest importance to mitigate future pressures (Veste et al, 2024; Ziervogel et al. 2014).*

*Line 27: ..........human needs with a more considered natural resource management approach something of importance, particularly in South Africa (Sheppard et al., 2020, Zerihun, 2021).*

2. All sections on data processing and quality control: I am missing a more detailed description on how e.g. the meteorological and soil data were filtered, using e.g. upper and lower limits (and which) or other despiking routines, such that the user knows about the quality of the data. Here you mainly report on wind speed and relatively vague on other parameter, but often not specific enough.

*For the most part the reported data is raw data where we only removed very obvious erroneous values that are attributable to sensor/logger failure etc For the variables where we did pre-processing, we explicitly state so in the data processing descriptions. We re-checked the files in this regard.*

3. I am aware that the data are linked to the paper of Hoffmeister et al. (2023) and that some data are presented there. From my perspective it would be nice to include a results section, where you present timeseries of the relevant parameter mentioned, such that I as an intended data user get an idea on how the data look like and about their quality. This would later on also support the "Application" section, where you can link your data/figure and the suggested future applications.

*Thank you for the suggestion. We agree that on overview of the data will help the readers to assess the possible applications and might increase usage of the data. We now included figures of the time series of the meteorological and hydrological variables. We did not put them in a separate "Results" section though, firstly because it's quite uncommon in a data paper and secondly because it's more a visualization of (almost raw) data and not a result of any analysis. We put the figure of the hydrological data in the respective data section, the one for meteorological data in the appendix as it is more of a general overview of ranges of the data.*

4. I also had a look on the data and found some errors. In parallel I checked the document "2023-028_Hassler-et-al_Data_Description.pdf", which is not always in line with the data found in the files. I am wondering, whether the published data are raw data or whether these are filtered? Otherwise I suggest checking the data again and updating the description file. From a user perspective this would be of high value.

*Thank you very much for also checking the data files and the consistency between description, data and manuscript. We re-checked the files to make sure they properly correspond to each other. We responded to your detailed comments about these files below.*

**Details on the files are:**

*2023-028_Hassler-et-al_Meteo_MeteorologicalMeasurements.csv:*

i) Wind velocity (Wind_v) is not in m s $^{-1}$, the magnitude is off and especially at the end of 2020 and whole 2021 there are many spikes, which might correspond to the filtered data for wind speed > 30 m s$^{-1}$
   *The values are indeed wind velocity in m s$^{-1}$, however, as we state in the data processing section, the sensor can have issues with moisture and then shows unrealistic values. We went through the data once more to find and remove all of them.*

ii) Gust_v: values of 30 m s$^{-1}$ at the end of 2020 and whole 2021 are not realistic and seems to be not filtered
   *We also went through the gust data one more time and removed any questionable values.*

iii) MaxPrecipRate: In the document it says mm h $^{-1}$ and in the paper it says "maximum rate for each 10-minute interval." So what is the truth?
   *Thank you for pointing this out, it was phrased ambiguously. The ATMOS41 weather station measures precipitation in individual droplets to reflect also dew formation with high accuracy and queries appropriate accumulations once every minute in memory. Precipitation is then given as cumulative amount according to a chosen interval and as a maximum rate for the respective interval multiplied by interval/60 min/h. We changed the text in the paper accordingly.*

2023-028_Hassler-et-al_Hydro_SoilMoistureMatricPotential.csv:

    i)       the soil water content is in % and not in $m^3\,m^{-3}$
               *Thank you for noticing this. We corrected the units in the revised data description.*

    ii)      spikes in ThetaWB60, ThetaWB80, ThetaWB10, ThetaB20, ThetaB40, ThetaB60, ThetaB80, TmatPotWB10, MatpotWB30, MatpotWB40, TmatpotWB40 should be removed
               *We removed some more spikes of obvious erroneous measurements and updated the data file. However, we did not do extensive data pre-processing, we only removed the values that we can definitely attribute to sensor/logger faults etc. All other values that are plausible remain in the data.*

    iii)     I am wondering why the matric potential is always positive, shouldn't it be negative, as you also state in the paper at l. 175?
               *Yes. The numbers are positive in the data file (which is a common way to report them), but matric potential is always negative. We attempted tol clarify this in the data description by putting "absolute values of matric potential" in the respective table.*

**Technical corrections:**

l. 12: The sentence sound weird: "The provided data is intended to explore the interaction between trees and crops in agroforestry landscapes."

*Maybe better:*

The provided dataset can be used to explore the interaction between trees and crops in agroforestry landscapes.

And I would also add an additional sentence as outlook on the potential use of the data set for different user, similar to what you've done in section 4 'Applications', but just summarised.

*Thank you for the suggestion, we changed the text as follows:*

*"The provided dataset provides a multidisciplinary approach to assess the impact and interaction of windbreaks and tree structures in agroforestry landscapes, aiding future work concerning water fluxes, nutrient distribution, microclimate, and carbon sequestration. The dataset, including high-resolution time series and point cloud data, offers valuable insights for managing the windbreak's influence and serves as a unique training dataset for spatial analysis."*

l. 29: please include a reference after this sentence: "By applying...it is demonstrated that AFS …. on an equivalent land area (e.g. … citation)."

*We modified the sentence before according to your previous comment (Specific comment #1), and put the sentence on the land equivalent ratio with the respective citation (Ong and Kho, 2015) in a separate sentence. This should clarify this passage.*

l. 30: same here, include reference please

*This sentence still refers to the LER concept by Ong and Kho (2015). However, as the citation is added already in the previous sentence we would consider the concept sufficiently cited and not cite the same paper again in this sentence.*

l. 40: separate sentences: … (Hombegowda et al., 2016). The presence…

*We changed this in the revision.*

l. 44: The question is, which temperature and humidity you refer to; I guess air temperature and humidity, if so, please write this, e.g.: … both air temperature and *air* humidty…

*We corrected this in the revision.*

l. 47: plural: … in agricultural *fields*

*We corrected this in the revision.*

l. 76: … focused *on* two …

*We corrected this in the revision.*

Caption Figure 1: rewrite caption: Images from the field site taken in a) winter and b) late spring.

*We changed this in the revision.*

l. 85: remove second *and*

*We corrected this in the revision.*

l. 86: which manual measurements of the windbreak trees were done? Have you measured the BHD or what else? Clarify please.

*We measured diameter at breast height (DBH) and between-tree spacing in order to cross reference the point cloud data. We added this information in the revision, in the tree measurement section 3.4.*

l. 91: include city and country of company: (METER Group,?,?) and also here ZL6 cloud Data Logger (company?, city?, country?)

*We added this in the revision.*

l. 93-108: I suggest to place the list of variables into a table. This would be easier to follow. You could then include the range, the resolution, the accuracy and the unit each in one column. And also include all other instruments described in the text and then include the company, city and country. This would be consistent.

*Thank you for the suggestions. We put the measured parameters, instruments, companies, range, accuracy and resolution in a table, one table for the meteorological measurements and one for the hydrological time series. For the laboratory analyses it does not really seem to be useful. The units of the measurements are reported as part of the detailed description of the data tables in the "Data description" file at the repository.*

l. 94: is 1750 W m $^2$ the maximum possible value the sensor can $^-$ measure or is this also the value you used to filter the data for outlier? If the latter applies keep in mind that global radiation can be max 1360 W m$^{-2}$.

*The value of 1750 W m$^{-2}$ is indeed the maximum possible value the sensor can measure which, as you rightly pointed out, is quite high given that the solar constant is 1361 W m$^{-2}$. We used the solar constant as a reference to filter the data for outliers, but fortunately all the values seem reasonable.*

Caption Figure 2: a point is missing at the end.

*We added it in the revision.*

l. 98: what are the values in brackets behind 0-100% (0-1.0), indicate! How is the accuracy changing with temperature and humidity? Please indicate!

*The values in brackets refer to the percentage, which, as usual, is given as a value between 0 and 1 in the CSV file.*

*The accuracy of the measurement is slightly lower when the relative humidity is high. We added a short explanation to the manuscript to clarify this: "The accuracy of the humidity sensor is generally slightly lower in a high humidity environment as it is a temperature-dependent variable that shows large fluctuations with small changes in temperature, the higher the number of vapour particles detected. "*

l. 100: out of curiosity: what is the temperature of the humidity sensor? Is there a second temperature sensor, e.g. used to calculate VPD? Or is this rather the dewpoint temperature calculated out of RH?

*Yes, there is a second temperature sensor. The vapour pressure sensor is located behind a circular Teflon screen in the same housing as the sonic transducers and measures relative humidity and temperature in addition to computing vapour pressure.*

l. 122: comma missing: Additionally, …

*We added it in the revision.*

l. 124:

→ as mentioned already, maybe include the sensor information into one common table.

*Yes, we put sensor information in tables, however, separately for the meteorological and hydrological time series.*

→ Here also be consistent in how you separate sensor name and company, use a comma instead of a semicolon.

*Thank you for noticing. We checked for consistency in the sensor/company references in the revised version.*

→ Also rewrite this sentence, e.g.: Soil moisture was monitored continuously … accuracy: +- 2%) in two profiles from … . If it is a profile, maybe directly mention the different installation depths in brackets behind.

*We adapted the first sentence in the revision. As the "profile" refers to the profile probe in the acrylic tubes which we describe in the following sentences, we agree it might be confusing. We changed the text to "monitored continuously… at two locations in four different depths, from…".*

→ from my point of view TDR is also an abbreviation, which you also used in Fig. 3, so please once write TDR out completely and abbreviate in brackets behind.

*We added the full name for the time domain reflectometry in the revision.*

l. 128: include comma: In each tube, …

*We added this in the revision.*

l. 149-152: I guess values in mm and μm refer to particle diameter, write this maybe like dp=…. With dp the particle diameter

*Thank you for the comment. We added the term particle diameter (dp) as you suggest to make it clearer.*

l. 153: include country (UMS GmbH, Munich, Germany).

*We added this in the revision.*

Figure 4: Write θ instead of Theta, this is more consistent with Eq. (1).

*We adapted this in the revision.*

l. 165: please include units for the different water contents or state that the soil water content is in either m³ m⁻³ or %.

*Thank you for noticing this. We harmonized the units in the revised data description.*

l. 173: what does this standard soil calibration refer to? Is the soil density of 1.40675 g cm⁻³ used in an equation to convert from e.g. raw voltage/resistance into soil water content? If an equation is used I suggest including it here.

*The devices' output is directly in vol. WC [%]. The conversion from electrical raw values to soil moisture are done internally, assuming by default a soil density of 1.40675 g cm⁻³. For details, readers are referred to the manufacturer's manual, which also offers suggestions to account for density deviations.*

Figure 5: → … according to *the* IUSS …

*We added this in the revision.*

→ reference to sub-figures a) and b) is missing in the caption

*We added the details for the sub-figures in the revision: a) Profile with the delineated horizons, b) profile pit in between the berry rows.*

l. 202 ff: split sentence in two and rewrite the sentence, this sounds weird. One suggestion would be:

The samples were homogenized, air-dried and sieved (<2 mm), and two sub-samples of each sample were transferred to Germany for laboratory analyses. Carbon and nitrogen contents *were estimated* at the Soil Ecology laboratory at the chair of Soil Ecology at the University of Freiburg, Germany, and soil texture was *estimated* at the laboratory of the research area Landscape Functioning at the Leibniz Centre for Agricultural Landscape Research (ZALF) in Müncheberg, Germany.

*Thank you for the suggestion, we changed these sentences to your version in the revision.*

l. 213: include city and country of company (Siebtechnik…)

*It's Siebtechnik TEMA, Mülheim an der Ruhr, Germany. We added it in the revision.*

l. 214: include comma: … dried at 105°C, combusted …

*We added it in the revision.*

l. 220: delete "two methods"

*We corrected this in the revision.*

l. 222: … University of Freiburg, Germany.

*We included the location as suggested (Germany).*

l. 226: These sieved particles were dried at 105°C [space] and the *sand fraction* [check missing text] was calculated …. .

*Thank you. We corrected this in the revision.*

l. 229: … particle size analysis *presented* in …

*We corrected this in the revision.*

l. 233: remove space between bracket and point.

*We corrected this in the revision.*

l. 234: which external standards were used?

*We used the Certipur, ICP multi-element standard solution IV (Merck KGaA, Darmstadt, Germany). Additionally we measured an internal soil standard with known nutrient contents together with the samples.*

*We added this information to the manuscript.*

l. 236: include city and country of company behind LiDAR (city, country).

*We added this information: (RIEGL Laser Measurement Systems GmbH, Horn, Austria)*

l. 241: include reference to software and what is the standard protocol?

*We added the software information which is by the same company as the instrument: (RIEGL Laser Measurement Systems GmbH, Horn, Austria). Also, we exchanged the "standard protocol" with the more detailed explanation: ... and following the software guided-steps to coarse register point clouds in an outdoor non-urban scenario, while using multi-station Adjustment 2 for fine registration in the same software.*

l. 245: specify where this video is included? E.g.: A video animation is included to the data set for …

*We added "…is included with the data on the repository for…" in the revision.*

l. 255: Split and rewrite sentence: The *studied* windbreak section was extracted from the co-registered project point cloud *and* filtered by pulse deviation (equal or lower than 10; Pfennigbauer and Ullrich, 2010). The isolated points were removed and exported as a separate file.

*Thank you for the suggestion, we included your text improvement, slightly adapting the last sentence: "Isolated points were removed and the windbreak point cloud was exported as a separate file."*

l. 266-267: What do the values in brackets refer to? What is the unit? Diameter maybe m?

*Yes, the values are  in metres. We added this directly after "Tested parameters were (in metres).*

l. 272: (>2mm) what do this refer to? Is it root diameter? If yes, please indicate.

*This refers to cross-sectional diameter. We adapted this to "(roots with > 2 mm cross-sectional diameter)" in the revision.*

Figure 6: include indication on position of sub-figure in caption, e.g.: Bird's eye *(top)* and side view *(bottom)* of …

*We changed this as suggested to:*

*"Figure 6. Bird's eye (top) and side view (bottom) of cylinder models of the sample windbreak. Colour denotes individual trees. Trees 1—4 (t01 to t04) were excluded from the analysis due to the windbreak edge influencing growth form."*

l. 288 and l. 289: include reference behind the R-packages

*We added these in the revision.*

l. 306: it is not crops your are referring to, aren't these shrubs or better just berry bushes? I think "… nutrient distribution and microclimate on the berry bushes." is better suited.

*We agree that it is not a "typical" annual crop but a perennial one. However, the term is wide and can even be applied to trees in forests or for biomass. We therefore left this unchanged.*

l. 306-312:

→ Here you seem to refer to results from Hoffmeister et al. (2023). Present these results in past tense.

*We corrected this in the revision.*

→ l. 306: By combining… This paragraph sounds out of place without a figure. From my perspective it doesn't make sense without presenting the results or at least some time series. Maybe it would be wise to include a results section in addition to/ before the Applications section?! This would also be different from results presented in Hoffmeister et al. (2023) and would give a quick overview on how the data look like. See also my comment at the beginning.

*We added overview graphs of the time series so that the reader can quickly grasp what the data looks like. However as it is not strictly a result from an analysis and rather visualization of data, we refrained from introducing a results section. We added time series graphs for important meteorological variables to the appendix and the ones for soil moisture/matric potential to the respective hydrology section.*

l. 317: TLS not defined: write out and abbreviate and "scans" can be deleted, as this is included in TLS

*We replaced this by "The availability of the point cloud data acquired with terrestrial LiDAR facilitates the three-dimensional examination of the studied windbreak in high detail…"*

References:

Veste, M., Sheppard, J. P., Abdulai, I., Ayisi, K. K., Borrass, L., Chirwa, P. W., Funk, R., Kapinga, K., Morhart, C., Mwale, S. E., Ndlovu, N. P., Nyamadzaw, G., Nyoka, B. I., Sebola, P., Seifert, T., Senyolo, M. P., Sileshi, G. W., Syampungani, S., and Kahle, H.-P.: The Need for Sustainable Agricultural Land-Use Systems: Benefits from Integrated Agroforestry Systems, pp. 587–623, Springer International Publishing, Cham, https://doi.org/10.1007/978-3-031-10948-5_21 , 2024

Zerihun, M. F.: Agroforestry Practices in Livelihood Improvement in the Eastern Cape Province of South Africa, Sustainability, 13, https://doi.org/10.3390/su13158477 , 2021.

Ziervogel, G., New, M., Archer van Garderen, E., Midgley, G., Taylor, A., Hamann, R., Stuart-Hill, S., Myers, J., and Warburton, M.: Climate change impacts and adaptation in South Africa, WIREs Climate Change, 5, 605–620, https://doi.org/10.1002/wcc.295, 2014

**Response to review 2 to Hassler et al.: "Multivariate characterisation of a blackberry-alder agroforestry system in South Africa: Hydrological, pedological, dendrological and meteorological measurements"**

Original review comments in black

*Answers in blue italics*

The data paper of Hassler et al. stems from an intensive field campaign in which hydrological, pedological, dendrological and meteorological measurements were made in an agroforestry system in South Africa.

The data collection and manuscript are both at a very high level. The manuscript is written and the data is will presented. The only limitation I see is that the paper/dataset has a very limited geographical scope, and hence re-use of the dataset may be limited too. I would honestly just have published this dataset as part of the accompanying paper Hoffmeister et al. (2023) instead of pursuing a separate data paper.

*Dear reviewer,*

*Firstly thank you for taking the time to review our manuscript. We are very happy to hear that you consider it to be of high quality.*

*We acknowledge the limited geographical scope but would like to point out the potential applications for the data set when placed in such an accessible location such as ESSD.*

*Some of the data was indeed utilised within Hoffmeister et al. (2023), however the ESSD paper includes more measurements than were analysed for that study, and the data can be used for a range of different other analyses: 1) The multidisciplinary measurements offer possibilities to study interactions of different ecosystem compartments that are otherwise rarely possible in discipline-specific campaigns. 2) The studied example of a (for the region typical) agroforestry system including windbreak and crop can be compared to other windbreak systems as well as to other agroforestry systems. This enables further insights into the varied interactions between the different components of these systems to help elucidate benefits and limitations of introducing agroforestry as a sustainable landuse option in a changing climate. 3) The detailed description of the individual (albeit small) disciplinary datasets supports further testing of methods and specific analyses, e.g. openly providing one of the rare point cloud datasets from terrestrial laser scanning for the trees of the windbreak, or testing feasibility of root water uptake calculations in such a setting.*

*These further applications support a detailed presentation of the data in ESSD and highlight their relevance, rather than just as a supplement to one study. We already emphasized some of them in section "4. Applications" of the manuscript, and we expanded it accordingly to show the potential use of the dataset.*

*Similarly, an ESSD paper also gives space to discuss limitations of the dataset in more detail compared to a mere supplement. We also expanded the applications section in this respect to make use of this possibility. For example, one aspect that we included here (and not only in the site description) is the irrigation scheme that was applied to the crops as this will influence for which purposes the data can be used.*